CellPress

## Perspective

# Genome annotation: From human genetics to biodiversity genomics

Roderic Guigó[1,2,*]

[1]Bioinformatics and Genomics, Center for Genomic Regulation (CRG), The Barcelona Institute for Science and Technology (BIST), Dr. Aiguader 88, 08003 Barcelona, Catalonia
[2]Universitat Pompeu Fabra (UPF), Barcelona, Catalonia
*Correspondence: roderic.guigo@crg.cat

### SUMMARY

Within the next decade, the genomes of 1.8 million eukaryotic species will be sequenced. Identifying genes in these sequences is essential to understand the biology of the species. This is challenging due to the transcriptional complexity of eukaryotic genomes, which encode hundreds of thousands of transcripts of multiple types. Among these, a small set of protein-coding mRNAs play a disproportionately large role in defining phenotypes. Due to their sequence conservation, orthology can be established, making it possible to define the universal catalog of eukaryotic protein-coding genes. This catalog should substantially contribute to uncovering the genomic events underlying the emergence of eukaryotic phenotypes. This piece briefly reviews the basics of protein-coding gene prediction, discusses challenges in finalizing annotation of the human genome, and proposes strategies for producing annotations across the eukaryotic Tree of Life. This lays the groundwork for obtaining the catalog of all genes—the Earth's code of life.

### INTRODUCTION

The biological traits of organisms are largely encoded in their genomes, specifically in defined genomic regions, the genes. Genes occupy a small fraction of most eukaryotic genomes (about 5% in the case of the human genome).[1] Identifying and mapping genes into a given genome sequence is usually referred to as annotating the genome. Annotating genomes is not a trivial task, as illustrated by the fact that more than 20 years after the completion of the first drafts of the human genome, the exact number of human genes is still unknown.[2] Projects are underway to sequence the genomes of all eukaryotic species known to live on Earth.[3] These are arguably among the most important projects in the history of biology as they will make it possible to identify the genetic events underlying the emergence of eukaryotic phenotypes. The value of genome sequences, however, is limited without an accurate annotation of the genes and the transcripts that originate from them since it is through the genes that we can link the genome sequence with the biology of organisms.[4]

While the concept of the gene is controversial,[5] genes can be operationally associated to transcriptional units, i.e., loci in the DNA sequence of the genome transcribed to mRNAs or other functional RNAs. Mapping or identifying genes in genomes is currently understood as identifying the location of functional RNAs within the genome and inferring their nucleotide sequence.

Identifying genes by sequencing the transcriptome (the set of all RNAs encoded into a given genome) is challenging. For a number of reasons, sequencing the transcriptome is more difficult than sequencing the genome. These reasons include the highly dynamic range of RNA abundance within the cell; the cell type and/or temporal specificity of RNAs; the heterogeneous localization of RNAs within the cell, with different RNAs species associated with different subcellular structures; the variable half-life of RNAs; and the complex steps in the pathways leading from DNA to mature RNAs. As a consequence, transcripts originating from the same genomic locus can co-exist simultaneously in the cell in different post-processing status (unspliced primary transcripts, spliced RNA precursors, small RNA products, …) and may carry out different biological functions. For instance, about one-third of small RNAs reside in protein-coding or long noncoding RNA (lncRNA) precursors.[6]

Thus, most methods to identify genes in genomes combine RNA sequencing, when available, with *ab initio* and comparative computational methods (Figure 1). *Ab initio* methods integrate known sequence biases in protein-coding regions that arise from the unequal usage of codons[7] and the sequence motifs typically found at splicing sites and translation initiation sites.[8] Comparative methods rely on the fact that protein-coding sequences tend to be more evolutionarily conserved than those that are not translated into proteins.[9] Thus, regions in the genomic sequence that exhibit sequence similarity with known proteins or with the genomes of other species are more likely to code for proteins.

Computational methods, however, have limited accuracy,[11] and early efforts were initiated to obtain full-length sequences for all human genes.[12] This was technically complex, as it required isolating the RNAs originating from a given locus, copying the RNA into complementary DNA (cDNA), and fragmenting it prior to sequencing. As an alternative to the costly

**Figure 1. Methods to predict genes in genomic sequences**

To produce the gene annotation of the genome—in the center of the figure, boxes correspond to exons and the blue segments to the coding regions—different sources of information are usually combined. The most informative is the mapping of RNAs sequenced from samples from the species (top). However, it is usually difficult to exhaustively survey all biological types within a species, and moreover, sequencing technologies have difficulties capturing low abundant transcripts and producing full-length sequences. Thus, mapping of RNAs is combined with computational methods. Within these, comparative methods identify regions in the genomic sequence that show similarity to known protein sequences (red profile) or directly to other genome sequences (green profile). "*Ab initio*" methods, on the other hand, use deviations in the composition of the nucleotide sequence characteristic of coding regions (coding potential) and predictions of splice sites and eventually other biological signals, such as translation start sites. The coding potential (shown as a profile in the figure) is related to the likelihood of a given genomic region to code for proteins based on codon usage (the deeper the valley in the figure, the more likely the region to code for proteins). A common method to predict splice sites is by means of position weight matrices (PWMs). The coefficients in these matrices are the observed probabilities of occurrence of each nucleotide at each position (which correspond to the sizes of the letters in the logos in the figure, generated using WebLogo).[10]

production of cDNAs, methods were also developed to simultaneously isolate tens of thousands of mRNAs and to sequence their 5′ and/or 3′ terminal ends.[13] Mapping these expressed sequence tags (ESTs) or full-length cDNAs to genomic sequences provides direct evidence for the existence of genes as they are proof of transcription. Moreover, it helps identify the 5′ and 3′ gene untranslated termini, which play an important regulatory role. These regions are more difficult to predict computationally because they do not show the composition biases of, and are less conserved than, protein-coding regions.[14]

EST sequencing produced the first genome-wide picture of the transcriptional complexity of the human genome. Initially puzzling, estimates of the number of human genes utilizing EST data (ranging from 60,000[15] to about 120,000[16]), were larger than estimates from protein-coding sequence conservation (28,000 to 34,000[17]). This discrepancy, however, was only apparent because these estimates were unknowingly counting different molecular entities. Sequence conservation-based methods[17] counted the number of protein-coding genes, while EST-based methods, depending on the sequence similarity threshold used to cluster ESTs into unique transcripts, were counting both protein-coding and non-coding genes[15] or all the alternative coding and non-coding transcripts encoded within these genes.[16] The paradox of the discrepant estimates of the total number of human genes, at the time of the first drafts of the human genome, illustrates how difficult it is to make sense of experimental data without the appropriate conceptual framework.

The development of massively parallel sequencing methods and their application to RNA sequencing (RNA-seq) extended and refined the picture of the transcriptional complexity of the human genome uncovered by EST sequencing, revealing that the number of lncRNAs is comparable to that of protein-coding

genes[18,19] and that alternative splicing is widespread—affecting nearly all human genes.[20] Early RNA-seq methods, however, produced sequences (reads) that are shorter than those of mature RNAs, making the assembly of complete transcript sequences from these reads challenging,[21,22] and had limited utility in annotating the human genome.

Technological advances have made the efficient sequencing of very long nucleotide molecules possible. When applied to RNA-seq, long-read methods, overcoming the limitations of short-read sequencing technologies, are able to produce sequences the length of entire transcripts and have been used to produce high-quality annotations.[23–27] In principle, long-read RNA-seq reads should be easily mappable to the genome and accurately resolve the exonic structure of transcripts. However, because of limitations—including library preparation protocols and the rate and nature of sequencing errors in current long-read technologies—sophisticated bioinformatic processing is still required to build transcript models from sequence reads.[28] This processing is not straightforward. An ongoing benchmark, the Long-read RNA-Seq Genome Annotation Assessment Project (LRGASP; https://doi.org/10.6084/m9.figshare.19642383.v1), shows that even with the input of the exact same long-read RNA-seq sequences, different bioinformatic pipelines can produce divergent results.

## CHALLENGES TO FINALIZE THE CATALOG OF HUMAN GENES AND TRANSCRIPTS

Computational methods, together with the many flavors of RNA-seq, and expert manual curation have been employed to identify human genes and transcripts.[29] Yet, despite enormous computational, experimental, and curation efforts and despite

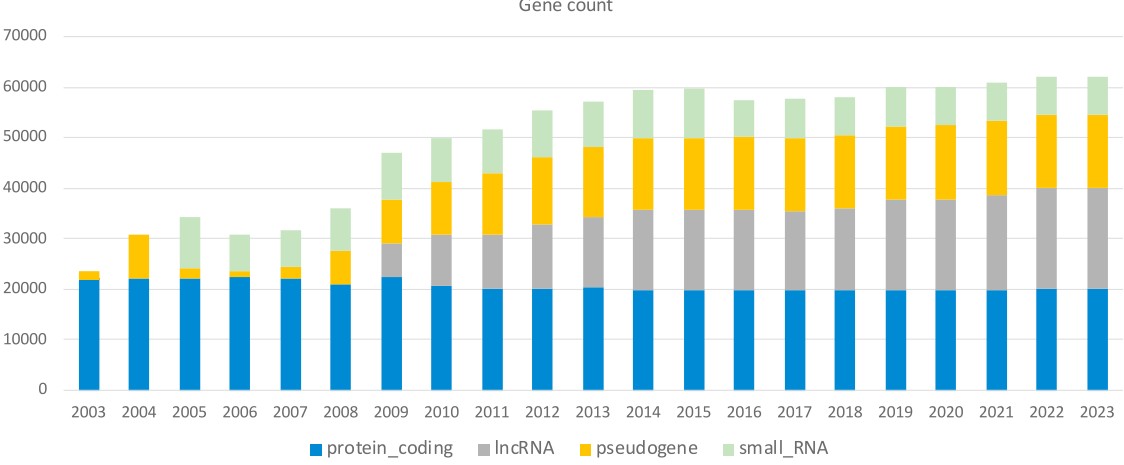

Gene count

Figure 2. Changes of the number of genes annotated in GENCODE over time

Number of protein-coding genes, lncRNAs, pseudogenes, and small RNAs annotated since 2003. lncRNAs were officially included in GENCODE in 2009. While the number of annotated protein-coding genes (and small RNAs) has remained stable (or even has slightly decreased) during the last two decades, the number of annotated lncRNAs (and, to a lesser extent, of pseudogenes) has steadily increased during this period. Small RNAs within GENCODE are predicted by fully computational pipelines, which were introduced in 2005. Changes in the numbers for small RNAs mostly reflect changes in the pipelines.

the availability of hundreds of thousands of transcriptional datasets,[30] the exact number of human genes is still unknown.[31] There are currently two main catalogs of genes and transcripts: GENCODE[29] and RefSeq.[32]

GENCODE currently has annotated about 60,000 genes corresponding to about 250,000 transcripts (Figure 2). Over 19,000 of these genes and about 89,000 of these transcripts are protein coding.[29] In addition to experimental evidence of protein-coding function, high-throughput proteomic experiments,[33] association with ribosomes (e.g., detected through RiboSeq[34]), and phylogenetic conservation across multiple genomes[35] are used in GENCODE as evidence to support the classification of a transcript as protein coding. RefSeq reports similar numbers of genes but only about 155,000 transcripts. In principle, the number of protein-coding genes encoding long open reading frames (ORFs) in the human genome is not likely to depart much from these estimates, and collaborative efforts have been initiated to converge upon a reference set of human protein-coding genes.[36,37] However, it remains unknown whether the human genome encodes a substantial number of small or non-canonical ORFs[38] coding for small peptides.[39] Demonstrating the coding capacity of small ORFs is challenging, as it is difficult to produce either multiple peptide evidence, as usually required as proof of translation, or strong signals of evolutionary conservation.

The number of human lncRNA loci, in contrast, is still unknown. The most recent estimates range from 19,000 in GENCODE to nearly 100,000 in NONCODE.[40] The discrepancies between these originate from the stringent manual curation in GENCODE, compared with NONCODE, which automatically includes RNA-seq-derived computational models. Unlike protein-coding transcripts, which have codon biases that skew nucleotide distributions, the sequence of long non-coding transcripts exhibit weak or no compositional bias, with typically little evolutionary conservation.[18] Therefore, lncRNAs can be identified mostly through RNA-seq. Since lncRNAs, as well as many alternative transcripts in protein-coding loci, are often expressed at low levels and in a cell-type-specific manner, it is hard to detect them using bulk RNA-seq. This is particularly true for experiments utilizing long-read RNA-seq given its comparatively low sequencing depth.[41]

Single-cell long RNA-seq methods,[42–45] which, in principle, exhaustively profile the cellular diversity of tissues and organs, could overcome the limitations derived from cellular specificity and low expression levels. However, challenges remain for RNA-seq methods to capture the full transcriptional complexity of human cells into a definitive catalog of genes and transcripts. First, prior to sequencing, RNA needs to be extracted and processed to produce the RNA (or cDNA) libraries that will be effectively sequenced. Ideally, these libraries should faithfully represent the RNA content of cells. However, many factors bias the representation of transcripts in sequencing libraries, including RNA processing status and post-transcriptional modifications, transcript length, cellular localization,[6] structural features,[46] etc. Specific protocols need to be employed to specifically target different RNA fractions. For instance, size selection is often employed to select transcripts in a given length range for sequencing.[47] The threshold between long and short RNAs is often arbitrarily set to 200 bp. There are a plethora of "short" RNA classes, some of which are well characterized, such as tRNAs, microRNAs, etc., with others less well studied, including piwi-interacting RNAs (piRNAs)[48] and the promoter associated RNAs known as PARS,[49] among others.

Moreover, the preparation of libraries for RNA-seq requires the isolation of the cells from their natural environment and/or their destruction, which may impact the transcriptome in ways we cannot anticipate. While spatial transcriptomic methods can visualize RNAs *in vivo*,[50] most can only monitor a set of previously annotated transcripts. Furthermore, because biopsies in healthy human tissues and organs can be difficult to obtain, RNA-seq is often performed on surgically removed diseased tissue or post-mortem samples. The removal from or the death of

the organism impacts tissue transcriptomes,[51] and there may be tissue-specific transcripts that rapidly cease to be expressed and/or degrade after the organism's death.

Technologies for real-time *in vivo* multiomics characterization of all the cells within a multicellular organism—without interfering with the system—appeared, until recently, beyond our imagination. However, advances are being made toward whole-transcriptome recordings, in which RNA is extracted from cells within their environment[52] or does not require cell destruction.[53] These technologies will be essential to detect genes and transcripts that are only transiently expressed such as those expressed in embryonic development, those controlled by circadian or circannual cycles, or those that respond to external factors or internal cues. The ability to monitor individual cells naturally functioning within living tissue during the normal functioning of the organism would greatly enhance the capacity to identify these transcripts. Induced pluripotent stem cell (iPSC)-derived cell lines are also a promising tool for genome annotation. They mimic differentiation and developmental processes, and they can be used, in addition, to explore the impact on the transcriptome of environmental and genetic perturbations, as some genes or transcripts may only be active in response to external cues.

Second, current human annotations are biased toward transcripts present in individuals of European ancestry due to their skewed representation in genomic datasets.[54] Efforts need to be made to capture the diverse transcriptomic makeup of human populations. This includes the generation of transcriptome datasets, ideally obtained through long-read RNA-seq, in diverse human population panels to extend short-read RNA-seq performed in samples available through the 1000 Genomes Project.[55]

Obtaining a universal human gene catalog may be a never-ending task. Indeed, the genome of each individual encodes a slightly different set of genes and transcripts, as a substantial amount of the human genome is present as copy-number variants (CNVs)[56]; these repetitive sequences can contain more or less gene copies across individuals.[57] Thus, specific gene-containing CNVs may not only be unique to certain populations[58] but also may be unique to a single individual.

Finally, there are ontological challenges in completing the catalog of human genes and transcripts. The first arises from the concept of gene. A catalog is generated from identifiable entities that can be counted. However, the discretization of the genome information that is implicitly assumed in the concept of gene does not fully capture the complexity of genomic information. Genes have been long regarded as well-bounded, discrete entities—as described by the "beads on a string" metaphor. While this metaphor could still be largely applied to protein-coding genes, coding and non-coding transcripts can overlap extensively, often in complex arrangements with fuzzy boundaries that do not necessarily respect co-linearity with the DNA sequence.[59] Thus, transcripts may not be discrete countable entities but may rather form a transcriptional continuum and be intrinsically uncountable.

Furthermore, a catalog often implies classification and systematization. While this facilitates comprehension, it also obscures the biological role of transcripts. The distinction between genes and pseudogenes, which are generally considered non-functional copies of functional elements, may be partially artificial. Many pseudogenes are transcribed,[60] and some play

biological functions.[61] Similarly, the distinction between protein-coding and non-coding genes may be an oversimplification. Most protein-coding loci generate both coding and non-coding transcripts,[62] and vice versa; many lncRNAs contain potentially translatable ORFs.[38,63]

Rather than assuming a binary typology, there is a continuum along the axis from protein-coding to non-coding transcription. Transcripts that are efficiently translated, containing long ORFs, would be preferentially classified as protein coding, while inefficiently translated transcripts containing small ORFs would be classified as lncRNAs. These could actually represent the emergence of novel protein-coding genes that often originate from ancestral lncRNAs.[64] The same transcript could eventually play a dual role as both an lncRNA or a protein-coding gene depending on the cellular environment or context, as has been widely documented.[65]

## BEYOND HUMANS: THE ANNOTATION OF ALL EUKARYOTIC GENOMES

Humans are just one extant species among the approximately two million known to inhabit the Earth. As of the end 2022, the genomes of about 10,000 eukaryotic species, about 0.5% of those known, have been sequenced.[66] The Earth Biogenome Project (EBP)[3] and associated projects, most notably the Darwin Tree of Life,[67] were launched with the goal of sequencing the genome of all known eukaryotic species. Essential to the impact of the EBP on biology is the ability to accurately annotate the genes encoded in these genomes. Lessons learned from annotating the human genome will certainly inform about the optimal strategies to produce gene annotations across the entire Tree of Life. The challenge, however, will be to produce these annotations as accurately as possible with likely only a fraction of the resources allocated to the human genome.

Most genomes are currently annotated using pipelines that automatically integrate *ab initio*, comparative, and RNA-seq information[68–70] (Figure 1). As technology progresses, long-read RNA-seq will likely become the preferred informative source (https://doi.org/10.6084/m9.figshare.19642383.v1). However, producing deep long-read RNA-seq will be feasible in the foreseeable future only for a small fraction of eukaryotic species due to the high cost, the difficulty to obtain tissue samples from which to extract sufficient amounts of high-quality RNA for many species, and other factors. To maximize the benefit for genome annotation of phylogenetically restricted long-read RNA-seq datasets, species should be selected in strategic locations within the eukaryotic phylogenetic tree so that high-quality annotations obtained in the selected species can be accurately propagated to as many species as possible within the same taxa or phylogenetic "neighborhood."

It is also important that, for multicellular eukaryotes, RNA is extracted from biological samples that maximize transcript diversity. This would be optimally accomplished through generating whole-body single-cell long-read RNA-seq atlases (Levy et al.[71]), which can capture, in principle, most RNA species across cell types. To maximize benefit with limited resources, one could devise a hierarchical approach (Figure 3). At the top of the hierarchy, whole-body long-read RNA-seq cell atlases, the deepest and most

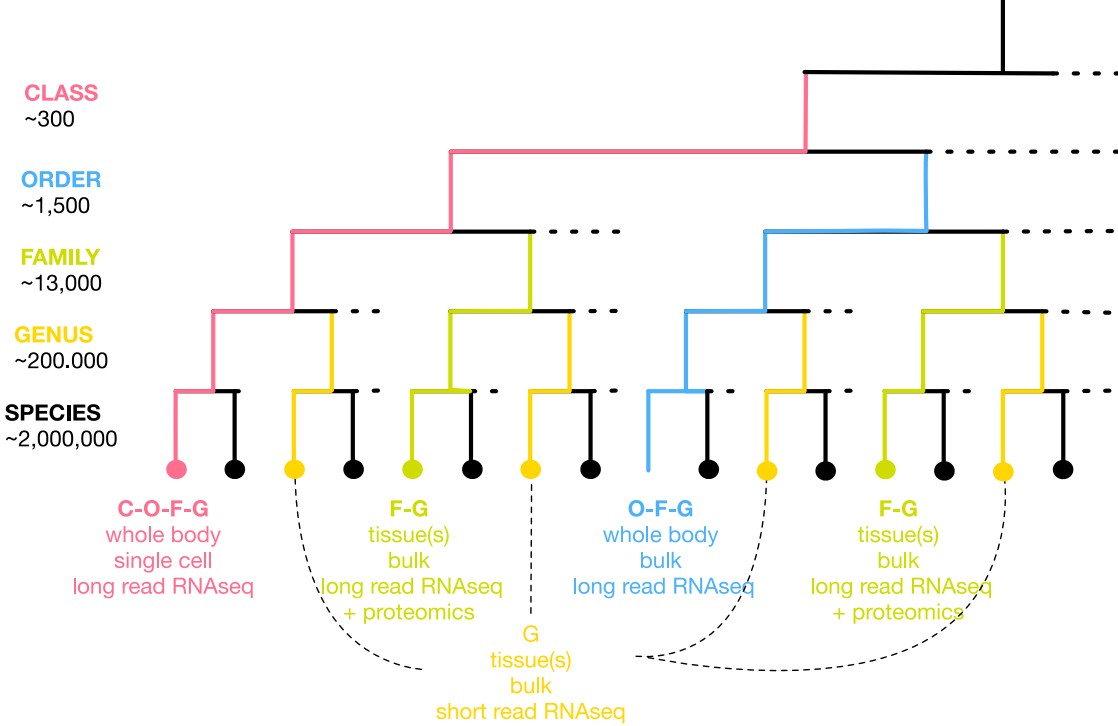

**Figure 3. RNA-seq to annotate all eukaryotic genomes on Earth**
Since resources are unlikely to exist to produce RNA-seq data for all species on Earth in the short term, species chosen for RNA-seq should maximize "annotation benefit," that is, to provide a framework to annotate, as accurately as possible, the genomes of as many other species as possible. A hierarchical approach would employ the most sophisticated (and expensive) RNA-seq methods in representatives of higher-order taxa and less sophisticated (and less expensive) methods applied to representatives of lower-order taxa (see text). In the figure, a cladogram represents a subset of the species within a class of multicellular eukaryotes. One species within the class is selected as a class representative (C, in red) and used for whole-body single-cell long-read RNA-seq. This species is also a representative of the order, family, and genus to which it belongs. For each of the remaining orders, a species is chosen as an order representative (O, in blue) for whole-body bulk long-read RNA-seq. This species is also a representative of the family and genus to which it belongs. Only two of the orders within the class are represented in the schema. For each of the families within each order, for which no species representatives exist as part of the class or order representatives, a representative is chosen (F, in green) for bulk long-read RNA-seq in selected tissues. These species are also representatives of the genus to which they belong. Finally, representatives are chosen for the remaining genus (G, in yellow) for short-read RNA-seq. Proteomics (eventually complemented with RiboSeq) could also be carried in species representatives at certain taxonomic levels (for instance, families). For protists, long-read RNA-seq (and eventually proteomics) could be carried in family representatives and short-read RNA-seq in genus representatives. Metagenomics combined with metatranscriptomics, which do not require species isolation and culture, could eventually become an alternative. Numbers for taxa are taken from Bánki et al.[72]

exhaustive transcriptome characterization, would be generated in one species representative of each of the approximate 300 eukaryotic classes. These atlases by themselves would constitute a unique resource to identify the eukaryotic cell types and establish their biological roles and evolutionary relationships. Then, at increasingly lower hierarchical levels, decreasingly deep and exhaustive (and less expensive) transcriptome experiments could be carried out. For instance, whole-body bulk long-read RNA-seq could be generated in one representative of each of the ~1,500 orders, tissue long-read RNA-seq in one representative of each of the ~13,000 families, and short RNA-seq in one representative of each of the ~200,000 genera. This taxonomy-based selection of representatives, however, may be suboptimal due to variable sequence diversity across taxa. Optimally sampling the eukaryotic sequence space may be preferable, although this sampling is difficult to determine prior to sequencing.

The dynamic range of gene expression poses a challenge to RNA-seq-based annotation. Unless resources exist to produce deep RNA-seq atlases, many lowly expressed transcripts will remain undetected. To access these transcripts in the human genome, methods have been developed that restrict long-read RNA-seq to targeted genomic loci[27] or that deplete sequencing libraries for highly abundant transcripts.[73] These methods, however, cannot be easily applied to the annotation of new genomes, as they depend on prior knowledge of transcript sequences.

Ideally, sequence-independent normalization methods that do not depend on previous knowledge should be developed. Some degree of normalization could potentially be achieved using nanopore sequencers, which can selectively sequence certain DNA molecules in a pool. This method could be used to avoid re-sequencing transcripts and to capture, at the same sequencing depth, a larger fraction of transcript diversity. In addition, sequencing libraries should be constructed to capture the entire length of the transcript (from the 5′ to the 3′), as the location of promoters is important for understanding gene regulation. A number of protocols are being developed toward that

aim. Such methods are generally based on selecting capped RNAs. These include teloprime[74] and technologies that combine long-read RNA-seq with CAP trapper protocol.[75] This protocol has been widely used for Cap Analysis of Gene Expression (CAGE),[76] which has been crucial to characterize the regulatory regions in the human genome.[77]

RNA-seq data, when available, will still need to be combined with computational methods to produce a catalog of protein-coding and non-coding transcripts. Methods would need to be developed to propagate high-quality annotations in representative species across the eukaryotic phylogeny. These could be inspired by current lift-over methods to map annotations across genome assemblies.[78] As the density of genome sequences, and of the corresponding annotations, grow across the eukaryotic phylogeny, one would expect that artificial intelligence methods would produce increasingly accurate annotation mappings across genomes with minimal human intervention.

In species for which many genomes are sequenced, such as humans, pangenomes[79] capture the genetic variability of the species more precisely than reference genomes. Methods will also need to be developed to annotate genes in the graph structure of pangenomes. This may be challenging for species with high genetic variability (which can reach up to 20% in eukaryotes),[80] as gene content may also be variable within species.

The need for standardized annotations across the entire Tree of Life argues for centralized pipelines and dedicated resources, such as those at Ensembl or NCBI. However, genome annotation is also a community task. Automatic pipelines, for instance, do not deal well with genes that depart from the canonical norm. These notably include selenoproteins, proteins that incorporate the amino acid selenocysteine in response to the UGA codon—a canonical stop codon,[81] and U12 introns, which often depart from the canonical GT-AG dinucleotide rule at intron boundaries.[82] Accurately annotating them often requires expert curation.

Expert curation may also be important to deal with other transcript types, which may be also challenging for automatic pipelines. This includes short, intronless, antisense, and chimeric transcripts and, in general, transcripts with low evolutionary sequence conservation, as well as pseudogenes. Expert community curation is particularly important for individual species and gene families to correct errors in automatic predictions and prevent them from propagating across genomes.[83] There is even an opportunity to engage the scientific community and even society as a whole in genome annotation—for instance, through student-driven community annotation projects.[84]

Developments in containerization software[85] and domain-specific languages (DSLs) to automate workflows[86] make it possible to implement, with limited effort, portable versions of the centralized pipelines producing highly reproducible results. These could be configured to take advantage of local domain expertise, leading to improved annotations while promoting scientific equity for data analysis.

Annotation pipelines will need to be continuously rerun for a given species, not only when new versions of the genome or new transcriptome data become available for that species but also as additional genomes for other species and their corresponding annotations are produced. This is because new and additional information can impact the annotation of any other species. There is a growing concern for the contribution of bioinformatics pipelines to the carbon footprint of humankind, particularly in the area of human genomics.[87] The projects to sequence the genomes of all biodiversity on Earth will produce an unprecedented amount of data. Therefore, analysis pipelines, including those for genome annotation, should be designed to most efficiently use computational resources.

## THE CATALOG OF ALL PROTEIN-CODING GENES ON EARTH

In the foreseeable future, exhaustive, multitranscript protein-coding and non-coding annotation supported by high-quality transcriptomic data will only be available for a small fraction of eukaryotes, hopefully strategically distributed across the eukaryotic phylogeny. Most genomes will be annotated by computational means, which produce acceptable annotations mostly for protein-coding genes. A catalog of these will still be a formidable resource. While transcriptional complexity is an intrinsic feature of eukaryotic genomes, one could easily argue that a large fraction of the biology of organisms is encoded in the set of protein-coding genes. Moreover, within protein-coding genes, often the same isoform is the most highly expressed across multiple biological conditions.[88,89] Thus, GENCODE v.41 annotates about 250,000 human transcripts, but only about 19,000 (8%) may correspond to the major protein isoforms.[2] This reduced representation of the human transcriptome explains a disproportionately large fraction of human biology.

Generating the equivalent of this reduced set of protein-coding transcripts for all eukaryotic species will maximize benefit given the limited resources. Efforts should be made thus to improve methods to predict protein-coding genes. This includes statistical methods to measure coding bias, in particular those that do not require known coding regions to be trained, methods for mapping protein to genome sequences at increasingly large evolutionary distances, and phylogenetic methods that better capture the characteristic features of protein-coding conservation.[35]

Methods to incorporate proteomics data in the gene prediction framework[90] should also be developed, as these data will be crucial to produce a complete catalog of the protein-coding genes encoded in a given genome. A program could be delineated, parallel to that outlined above for transcriptome data, in which proteomics is carried out in species strategically placed in the eukaryotic tree (Figure 3). To maximize benefit whenever possible given sample availability, proteomics data should ideally match transcriptomics data regarding the species and the conditions/tissues being monitored.

RiboSeq data can efficiently complement proteomics data by lowering the stringent threshold employed to consider mass spectrometry (MS) proof of peptide existence.[91] A community-led effort to bring RiboSeq, together with proteomics data, to annotate protein-coding function in human genes could serve as a model for other species.[34] These experimental approaches are essential to identify species- or taxa-restricted protein-coding genes, which are likely to escape detection through sequence comparative approaches but may play a very important role in the emergence of taxa-specific phenotypes.

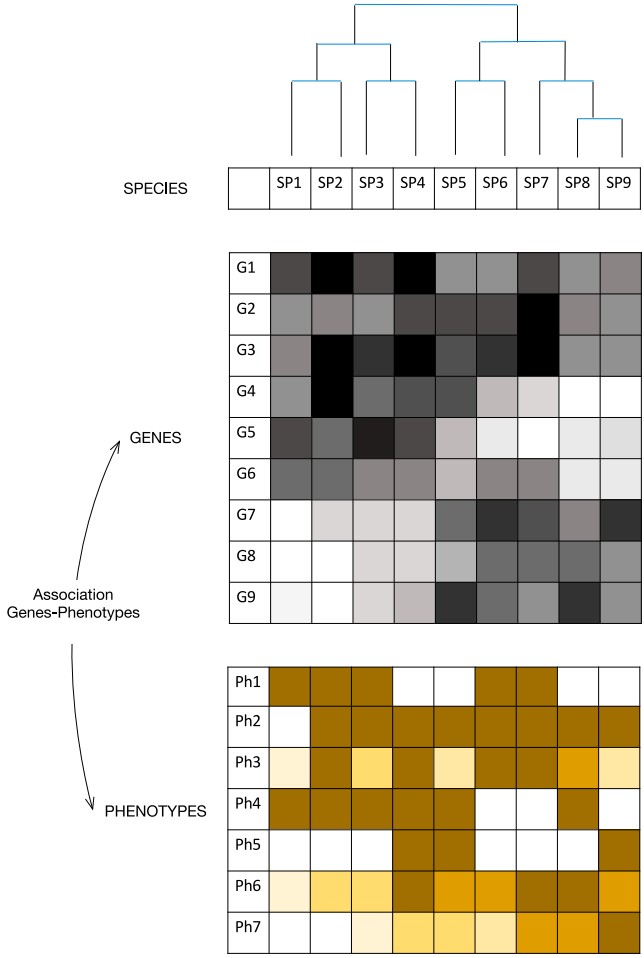

**Figure 4. The universal protein-coding gene set**
Once all genomes are annotated, the set of all protein-coding genes could be, in principle, derived from orthology across species. This is not a trivial task, as paralogy often obscures orthology relationships, and orthology should probably be established among clusters of genes.[96] Species could be defined by subsets of the universal gene set. Many genes (or clusters) will be present across all eukaryotic species, and many will be restricted to specific taxa or even species, but many will not necessarily reflect the eukaryotic phylogeny, as they may have originated independently (from the same orthologous loci) or selectively lost multiple independent times. The presence/absence of genes within a given species can be further quantified with reference to some archetypical sequence for each gene (denoted by shades of gray in the figure; the darker the cell, the more similar the sequence of the gene from that species is to the archetypical sequence for that gene; blank cells denote the absence of the gene in the species). The universal gene set (or subsets within) can be employed for genome-to-phenome associations across all eukaryotic species (or subsets within). The presence/absence of phenotypes in species (or variations in quantitative phenotypes, denoted by shades of brown in the figure) can be correlated to the presence/absence of genes (and/or to similarity to the archetypal gene sequences) in a vague analogy to GWASs.

## GENOME-PHENOME ASSOCIATIONS ACROSS THE EUKARYOTIC PHYLOGENY

Because of sequence conservation, the evolutionary history of protein-coding genes, unlike that of most lncRNAs and other genome elements, can be traced across species and the orthology established. A universal catalog of all eukaryotic (protein-coding) genes would thus facilitate a gene-centered view of biology in which genes are the unit of selection.[92] Species would be defined as subsets of this universal eukaryotic gene set, and genome-phenome studies could be carried out across up to hundreds of thousands of species, in which the presence/absence of genes (or sets of genes) or variations in the rates of their evolution could be associated with the emergence of phenotypes (Figure 4). These association studies would contribute to uncovering the dramatic genomic events underlying major transitions during the history of life, as well as those underlying the emergence of recurrent phenotypes via convergent evolution,[93] such as regeneration or sociality, or inter-specific differences in quantitative phenotypes, such as body size and longevity, among many others. Recent examples of the increasing power of comparative genomics to identify the genetic basis underlying the emergence of phenotypes are the sequencing the genomes of 240 placental mammals[94] and of 233 primates[95]

There is ample literature specifically regarding the association of gene expansions and/or losses with eukaryotic phenotypes. For instance, loss of entire gene families has been reported in bats, connected to changes in the DNA damage pathway, and linked to the origin of flight,[97] and pseudogenization of genes related to chemosensation has been reported in mammals adapted to aquatic environments.[98] Generic methodologies to associate gene losses/gains with phenotypes, however, have been preferentially developed in prokaryotes (see MacDonald and Beiko[99] and Feldbauer et al.[100] and references therein). Prokaryotes are a good model for such endeavors; studies in prokaryotes have an easier time identifying gene orthology (at least within a given taxa) and gene loss (because pseudogenization is not as common as in eukaryotes[101]), and there is abundant phenotypic data in machine-readable format.[102,103] A universal eukaryotic gene set would facilitate genome-phenome associations across tens of thousands species. This will require the generation of phenotypic ontologies across eukaryotes[104] in which equivalent phenotypes can be consistently labeled across species.

Methods will need to be developed to scale to large datasets. Inspiration could come from genome-wide association studies (GWASs), which are grounded in statistical theory and for which efficient methods have been developed. In GWASs, the genomes of multiple individuals within a species are compared to identify loci (usually single nucleotides) that segregate with specific traits. In a potential cross-species "comparative GWAS," species would play the role of individuals and genes (or other coarse genomic elements) the role of single nucleotides (Figure 4). The aim is to identify genes or sets of genes associated with a given phenotype. The GWAS analogy has its obvious limits. In most GWASs, both the tested genetic feature (i.e., a single-nucleotide variant) and the associated phenotypes are assumed to be homologous, with the same biological mechanisms underlying the association across the genomes analyzed. In comparisons across species, reliably establishing orthology between genomic features (even in the case of protein-coding genes, despite recent progress[105]) and between phenotypes may be challenging. Moreover, the biological mechanisms

connecting the genome with the phenome may not be conserved, as homoplasy can originate from different molecular mechanisms in different species.

Because of the strong impact of shared ancestry, any general framework for linking the genome to the phenome needs to be phylogenetically aware.[106] This is an additional challenge, as it is unclear how well phylogenetic methods scale to thousands of genomes. In particular, such analyses often depend on building multiple sequence alignments, which deteriorate with the size of the input sequence set.[107,108]

The Human Genome Project was reluctantly received by some individuals, who perceived it as a deviation of resources that could have been better invested in more traditional, hypothesis-driven, projects. Today, there is almost unanimity that the impact of the project has been extraordinary—much larger than optimistically anticipated. This impact may be dwarfed by projects to sequence the genomes of all (eukaryotic) species on Earth, as they will make it possible to connect the genotype with the phenotype at unprecedented scale. This impact will be felt beyond biology and will impact many aspects of the life sciences and technologies. Indeed, because of the interconnected nature of life, genomes across multiple species contribute to the understanding of the biology of individual species, such as humans. For instance, the patterns of sequence conservation of genes across multiple species help to assess the potential pathological impact of mutations in the human genome.[109]

Within genomes, phenotypes are mostly encoded in protein-coding genes. Because of sequence conservation, these can be used to establish phylogenetic relationships between genomes. Thus, to maximize the impact of sequencing the genomes of all Earth biodiversity, strategies need to be delineated to produce, with limited resources, an accurate catalog of all protein-coding genes in eukaryotes. This involves generating, in addition to genome sequences, functional genomics data (transcriptomics, proteomics, etc.) in species strategically located within the eukaryotic Tree of Life and developing computational methods to propagate accurate annotations in these species. Development of methods to predict genes that show less sequence conservation and constraint, such as lncRNAs, should not be forgotten, given that these genes also contribute to the final phenotypic outcome of genomes. The set of all (eukaryotic) genes on Earth will allow for a holistic vision of biology, as it is not possible to fully comprehend the biology of a given species without comprehending the biology of all.

### SUPPLEMENTAL INFORMATION

### ACKNOWLEDGMENTS

I would like to thank José González and Adam Frankish from the European Bioinformatics Institute for sharing the data in Figure 2 and all GENCODE colleagues for all the years working together. I would also like to thank the reviewers for constructive feedback. I would like to acknowledge the Catalan Initiative for the Earth Biogenome Project (Institut d'Estudis Catalans) for creating an environment that has engaged many of us beyond our specific area of expertise. Finally, I would like to apologize to the colleagues in my group for stealing time from their projects to write this perspective and to thank specially Romina Garrido for all the help these years. Core support for the research at the CRG is from the CERCA Programme/Generalitat de Catalunya and from the Spanish Ministry of Science and Innovation to the EMBL partnership, Centro de Excelencia Severo Ochoa.

### DECLARATION OF INTERESTS

The author declares no competing interests.

### INCLUSION AND DIVERSITY

We support inclusive, diverse, and equitable conduct of research.

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
