## [Document S1. Transparent peer review records for Guigó et al. · Cell Genomics]

Cell Genomics, Volume 3

Supplemental information

Genome annotation:

From human genetics

to biodiversity genomics

Roderic Guigó

Genome annotation: from human genetics to biodiversity genomics.

Roderic Guigó

Summary

Initial submission: Received : 12/14/2022

Scientific editor: Laura Zahn

First round of review: Number of reviewers: 3
Revision invited : 2/22/2023
Revision received : 4/25/2023

Second round of review: Number of reviewers: 3
Accepted : 7/7/2023

Data freely available: N/A

Code freely available: N/A

This transparent peer review record is not systematically proofread, type-set, or edited. Special characters, formatting, and equations may fail to render properly. Standard procedural text within the editor's letters has been deleted for the sake of brevity, but all official correspondence specific to the manuscript has been preserved.

Referees' reports, first round of review

Reviewer #1: This perspective proposes a possible strategy towards describing the diversity of genes and transcripts across the tree of life. The manuscript briefly introduces on the three pillars of gene annotation: ab initio methods, comparative methods, and transcriptome sequencing. An overview of the current state of human annotations is then presented, with an emphasis on transcript detection. Technical and conceptual challenges that currently prevent to have a complete list are presented, and it is suggested that such list may never exist based on an argument that genes and transcripts are intrinsically uncountable. Notwithstanding these challenges, the author proposes that the science community can nevertheless push towards annotating genes and transcripts throughout the tree of life. The approach suggested by the author would combine continuous genome annotation where gene findings in any species can inform all others with a phylogeny informed RNA sequencing strategy. After highlighting some areas of research that would make this vision possible, the author suggests that a focus should be placed on protein-coding genes and discusses how obtaining a universal catalog of protein-coding genes would inform a gene-centered view of biology where genes, not species, are the unit of selection. It is argued that such a vision, together with additional resource development, would enable to perform GWAS-like investigations of genotype-phenotype relationship at a scale never seen before, to investigate the genomes of ecosystems, and to trace the history of all protein-coding genes on earth.

The topic of this perspective is fascinating and indeed in my view a most important goal of the field of genomics. The part of this perspective describing the methods and challenges in transcript annotation of the human genome is very well written and informative. I do however have some serious issues with the general strategy that is proposed:

- A good argument is made to focus on annotating protein-coding genes rather than all transcripts across the tree of life. In this context, I don't understand why the author dismisses the approaches that directly detect protein-coding capacities of sequences (ribosome sequencing and proteomics) and proposes to focus on transcripts. This is even more surprising given that the perspective insists on the aspects that make transcript sequencing and ab initio sequence analysis not sufficient. While ribo-seq and proteomics have limitations, they both are better suited to protein-coding sequence detection than transcript analysis.
- The stated goal of the proposed strategy is to describe the diversity of genes on earth. However, the proposed strategy focuses exclusively on gene families and is therefore bound to miss species-specific and taxonomically restricted orphan genes, including all the rapidly evolving microproteins that are currently being discovered by unbiased ribosome sequencing analysis. As described, the approach would only yield annotations of protein-coding genes that belong to known gene families. This would already be a fantastic goal to reach in my view, and the author may want to revise his perspective to state that this is the goal and that the study of the myriad of additional genes and short translated elements that do not belong to gene families could be another enterprise that is taken on later. If the author really wants to propose a way towards a "catalog of all protein-coding genes on Earth", then orphan genes and microprotein encoding genes cannot be ignored. The strategy would need to be seriously revised to achieve this broader goal, and in my view that would be a very worthwhile effort since it is likely that taxonomically restricted protein-coding sequences influence taxonomically phenotypes through rapidly evolving mechanisms.
- The last part of the perspective, where a "gene-centered view of biology" and GWAS-like approach to unraveling genome-phenome relationships are described, is currently very insufficiently documented. For instance, an entire field of evolutionary research already focuses on linking genes gains and losses across phylogenies to phenotypes. Of course, the field deals with smaller phylogenetic scales currently as the tree of life is not yet annotated. Yet many methods are already developed and have yielded results exactly along the line of what the author proposes, beyond the handful of examples cited by the author restricted to microbes. A good example that uses a general framework for convergent evolution analyses and that has been applied for many phenotypes is described in *Science*. 2018; 361(6402): 591-594. I recommend that the author reviews this field in this section instead of seeming to propose it in the text and figure 4.

In addition to these major concerns, I find that the perspective touches upon many topics in passing without developing or including citations (particularly after Beyond Humans, some examples below), which makes the flow of ideas difficult to follow. I have additional comments:

- The proposed hierarchical approach to transcript sequencing is well described in the text and in my view very worthwhile, but Figure 3 is unclear. The numbers listed on the figure do not match those in the text p.6, and visually the choice of tree representation begs the question of why not propose all these transcript measurements to be done on one single species since it is a representative of all the tree nodes drawn here. The author may for example expand the tree representation to show two classes instead of one, each with the derived subgroups, so the scope of the concept becomes visually clear.
- The brief description of ref. (22) on p. 3 is misleading. While mere ribo-seq read mapping to transcripts is indeed only evidence of association with ribosomes and not a sufficient demonstration of translation, many methods have been developed to tease these two apart and detection of triplet periodicity in the ribo-seq reads is indeed evidence of translation (but not necessarily of the existence of a stable protein product resulting from this translation).
- The comparison between GENCODE and NONCODE on p.4 is very interesting and it would be great to explain what different "rules" lead to one number versus the other.
- What is meant on p.5 by "(partially) engineered" genomes, and how would this render personalized gene catalogs more prevalent?
- The argument on p.5 about "uncountable" genes and transcripts is interesting but seems flawed to me. The argument is well made, in particular in the following paragraph, why it could be argued that transcripts are uncountable. But how does this argument apply to genes? And, if genes are uncountable, then how does that impact the efforts to annotate genes across the tree of life?
- The notion of continuum between non-coding and coding, p. 5, was proposed in Nature volume 487, pages370-374 (2012).
- Figure 2 shows change in annotation status of different categories of genes over time in GENCODE, but the main text only talks about the current status, and briefly mentions an increase in annotations in 2009 in the captions. However, there are large fluctuations in protein-coding annotations prior to 2009, and there is the sudden emergence of small RNA annotations in 2005 that seem to be worthy of discussion from a historical context. Also, the "v" dates (e.g. 2005 versus 2005v) in the x axis should be clarified - what do they mean?
- Page 5 third paragraph: "transcripts actually overlap extensively, often in complex arrangements that do not necessarily respect co-linearity with the DNA sequence" - citations should be included.
- Page 5 fourth paragraph: "the same transcript could have a role as both lncRNA or protein coding gene in different conditions or in different cellular compartments" and "the dual function of many transcripts has been widely documented" - Citations to concrete examples of dual coding/noncoding transcripts would be nice (e.g. SPAAR/LINC00961; Spencer et. al 2020).
- Page 7 paragraph 1: "Ideally, sequence independent normalization methods should be developed." - What is sequence independent normalization?
- Page 7 paragraph 2: "Automatic pipelines, for instance, do not deal well with exceptions to the universal genetic code." There are other cases that automatic pipelines do not deal well with that may be good to note, such as the identification of shorter coding sequences.
- Page 9 paragraph 1: "Genetic relatedness is likely to have a much larger confounding effect in across-species than in within-species comparisons, as the genomic space is also much larger and, in consequence, it is the range of genetic similarities between genomes." - This sentence seems to have grammatical errors? I am not sure of the meaning but indeed the importance of phylogeny awareness in predicting genotype-phenotype relationships from multi species comparison is well documented.

Reviewer #2: Comments enter in this field will be shared with the author; your identity will remain anonymous. This overview is essential as there are many misconceptions about the process and difficulties of gene annotation. After the publication of the human genome, many researchers, including myself, thought that computational annotation would largely solve the gene annotation problem. In not human genomics circles, there is a strong emphasis on assembly, often with the assumption that annotation will be "free". This article brings the history, current challenges, and opportunities to a broader audience of researchers who are very much dependent on this work.

Discussion of issues around annotating TSS/TES, the value of CAGE, and other methods to identify full-length transcripts would enrich this article.

For Figure 2, it would be good to explain why the number of annotations went down in the early stages of

GENCODE. Discussing this will head off questions and help emphasize the importance of new sequencing data.

The text in several figures is fuzzy, indicating the use of bitmapped rather than vector graphics images.

Reviewer #3: Comments enter in this field will be shared with the author; your identity will remain anonymous.

In the manuscript, Dr. Guigo reviewed the history of gene annotation, the different strategies, and then discuss the challenges for finalizing gene annotation in human genome, as well as generating annotations across different eukaryotic species. Overall, the manuscript, if as a review, did not cover many of the subjects in sufficient depth, if as a perspective, did not provide enough novel insights. Therefore, I do not think it is suitable for a prominent journal as "Cell Genomics".

Authors' response to the first round of review

Reviewer #1: This perspective proposes a possible strategy towards describing the diversity of genes and transcripts across the tree of life. The manuscript briefly introduces on the three pillars of gene annotation: ab initio methods, comparative methods, and transcriptome sequencing. An overview of the current state of human annotations is then presented, with an emphasis on transcript detection. Technical and conceptual challenges that currently prevent to have a complete list are presented, and it is suggested that such list may never exist based on an argument that genes and transcripts are intrinsically uncountable. Notwithstanding these challenges, the author proposes that the science community can nevertheless push towards annotating genes and transcripts throughout the tree of life. The approach suggested by the author would combine continuous genome annotation where gene findings in any species can inform all others with a phylogeny informed RNA sequencing strategy. After highlighting some areas of research that would make this vision possible, the author suggests that a focus should be placed on protein-coding genes and discusses how obtaining a universal catalog of protein-coding genes would inform a gene-centered view of biology where genes, not species, are the unit of selection. It is argued that such a vision, together with additional resource development, would enable to perform GWAS-like investigations of genotype-phenotype relationship at a scale never seen before, to investigate the genomes of ecosystems, and to trace the history of all protein-coding genes on earth.

The topic of this perspective is fascinating and indeed in my view a most important goal of the field of genomics. The part of this perspective describing the methods and challenges in transcript annotation of the human genome is very well written and informative. I do however have some serious issues with the general strategy that is proposed:

I am glad the reviewer considered the topic of this perspective fascinating and a most important goal of the field of genomics. I also thank them for the detailed comments and criticisms, which I have taken into account in writing this revised version. I believe they have helped to improve the manuscript. I want to note that the general strategy outlined in the last section of my previous submission was more of a proposal than a detailed program, which I believe is beyond the scope of this perspective. Still, I have addressed the issues raised by the reviewer, and I have substantially re-written the last section of the proposal. This is now divided in three sections: "Beyond humans: the annotation of all eukaryotic genomes", where I discuss how the

lessons learnt from annotating the human genome can be used to address the challenges of annotation all eukaryotic genomes; “The catalog of all protein coding genes on Earth”, where I discuss the value of such a catalog and suggest approaches to generate it; and “Genome-Phenome associations across the eukaryotic phylogeny”, where I discuss the methodological challenges to use this catalog to identify genome-phenome associations. In addition, I have fully redrawn Figure 3, as suggested by the reviewer.

- A good argument is made to focus on annotating protein-coding genes rather than all transcripts across the tree of life. In this context, I don't understand why the author dismisses the approaches that directly detect protein-coding capacities of sequences (ribosome sequencing and proteomics) and proposes to focus on transcripts. This is even more surprising given that the perspective insists on the aspects that make transcript sequencing and ab initio sequence analysis not sufficient. While ribo-seq and proteomics have limitations, they both are better suited to protein-coding sequence detection than transcript analysis.

The reviewer is totally correct. This was really an oversight from my side. I have now included a discussion on the importance of proteomics and RiboSeq data towards the goal of building the catalog of all protein coding genes (paragraphs 3 and 4 on page 10). I have also mentioned explicitly proteomics data in the new version of Figure 3

- The stated goal of the proposed strategy is to describe the diversity of genes on earth. However, the proposed strategy focuses exclusively on gene families and is therefore bound to miss species-specific and taxonomically restricted orphan genes, including all the rapidly evolving microproteins that are currently being discovered by unbiased ribosome sequencing analysis. As described, the approach would only yield annotations of protein-coding genes that belong to known gene families. This would already be a fantastic goal to reach in my view, and the author may want to revise his perspective to state that this is the goal and that the study of the myriad of additional genes and short translated elements that do not belong to gene families could be another enterprise that is taken on later. If the author really wants to propose a way towards a "catalog of all protein-coding genes on Earth", then orphan genes and microprotein encoding genes cannot be ignored. The strategy would need to be seriously revised to achieve this broader goal, and in my view that would be a very worthwhile effort since it is likely that taxonomically restricted protein-coding sequences influence taxonomically phenotypes through rapidly evolving mechanisms.

Again the reviewer is correct. Proteomics (complemented with RiboSeq data) will be crucial to identify taxa-restricted protein coding genes, which may play a very important role in defining taxa-specific phenotypes. I explicitly mention this in the revised version of the manuscript: “These experimental approaches are essential to identify species or taxa-restricted protein coding genes, which are likely to escape detection through sequence comparative approaches, but may play a very important role in the emergence of taxa-specific phenotypes.” (paragraph 4 on page 10).

- The last part of the perspective, where a "gene-centered view of biology" and GWAS-like approach to unraveling genome-phenome relationships are described, is currently very insufficiently documented. For instance, an entire field of evolutionary research already focuses on linking genes gains and losses across phylogenies to phenotypes. Of course, the field deals with smaller phylogenetic scales currently as the tree of life is not yet annotated. Yet many methods are already developed and have yielded results exactly along the line of what the author proposes, beyond the handful of examples cited by the author restricted to microbes. A good example that uses a general framework for convergent evolution analyses and that has

been applied for many phenotypes is described in Science. 2018; 361(6402): 591-594. I recommend that the author reviews this field in this section instead of seeming to propose it in the text and figure 4.

In addition to these major concerns, I find that the perspective touches upon many topics in passing without developing or including citations (particularly after Beyond Humans, some examples below), which makes the flow of ideas difficult to follow.

As mentioned, I have reorganized and extensively edited this part of the manuscript. I believe that most concerns by the referee relate to this part, and specifically to the now last section of the manuscript “Genome-Phenome associations across the eukaryotic phylogeny”. I absolutely agree that using comparative genomics to identify emergence and evolution of traits is an entire, well established, field of research. My intention in this section is not to produce an exhaustive review of the field, but to emphasize that a universal catalog of protein coding genes will empower the field with the capacity to perform genome/phenome associations at a different, coarser structural level at a much larger scale. Comparative methods often employ models of molecular evolution to identify patterns of sequence variation, usually in a limited number of loci, that correlate with phenotypic traits. These methods typically compare dozens, maybe hundreds of genomes, but it is unclear how well they will scale to thousands, tens or even hundreds of thousands of genomes. In particular, they often depend on building multiple sequence alignments, which are known to deteriorate with the size of the input sequence set.

In this section, I specifically discuss that a universal catalog of all protein genes would facilitate the identification of genome/phenome associations in which the unit of variation is not the nucleotide, but the gene. I absolutely agree, as well, that “there is already ample literature on methods associating single genes or clusters of genes to phenotypes”, as I already acknowledged in the original submission. Many of the examples are indeed in eukaryotes, and I have included the Meyer et al (2018) reference on the adaptation to aquatic environments and the Zhang et al (2013) reference on the origin of flight in bats, among the many that could be included as examples. But I believe that, for the reasons explained in the manuscript, generic frameworks have been preferentially developed in prokaryotes (“studies in prokaryotes have easier to identify gene orthology (at least within a given taxa), with easier to determine gene loss (because pseudogenization is not as common as in eukaryotes (94)) and there is abundant phenotypic data in machine readable format (95,96).”)

A universal catalog of genes will facilitate the development of generic frameworks also in eukaryotes, as it will allow for unequivocally determining all gene losses in a given genome, which is not irrelevant. Here, but just as one among the many possible avenues, I borrow from the GWAS analogy, as GWAS are grounded in solid statistical theory, and for which efficient methods have been developed that do scale to tens or hundreds of thousands of genomes.

I believe that the new version of this section, in which I have included additional references within the constraints imposed by the journal guidelines, addresses most of the issues raised by the reviewer

I have additional comments:

- The proposed hierarchical approach to transcript sequencing is well described in the text and in my view very worthwhile, but Figure 3 is unclear. The numbers listed on the figure do not match those in the text p.6, and visually the choice of tree representation begs the question of why not propose all these transcript measurements to be done on one single species since it is a representative of all the tree nodes drawn here. The author may for example expand the tree representation to show two classes instead of one, each with the derived subgroups, so the scope of the concept becomes visually clear.

Thanks to the reviewer for this suggestion. I have re-drawn the figure according to it. I hope that

the reviewer finds this representation much more clear.

Regarding the number of taxa, I have now resorted to what I believe is the most authoritative source, the Catalogue of Life, <https://doi.org/10.48580/dfrq>. Numbers, especially for families, have changed some.

- The brief description of ref. (22) on p. 3 is misleading. While mere ribo-seq read mapping to transcripts is indeed only evidence of association with ribosomes and not a sufficient demonstration of translation, many methods have been developed to tease these two apart and detection of triplet periodicity in the ribo-seq reads is indeed evidence of translation (but not necessarily of the existence of a stable protein product resulting from this translation).

As mentioned above, I included a discussion on the importance of proteomics and of riboSeq data (paragraphs 3 and 4 on page 10). I do agree that Riboseq can efficiently complement proteomics data.

- The comparison between GENCODE and NONCODE on p.4 is very interesting and it would be great to explain what different "rules" lead to one number versus the other.

Essentially, GENCODE only includes manually curated data, while NONCODE includes RNAseq derived computational models. Many of these are not validated by the GENCODE curators. I comment on this in the revised version of the manuscript (paragraph 4, page 4).

- What is meant on p.5 by "(partially) engineered" genomes, and how would this render personalized gene catalogs more prevalent?

I removed this sentence. I meant that through genome editing, gene copies could be artificially included into the genomes of individuals in a personalized way. I do not think it is very relevant in the context of this piece.

- The argument on p.5 about "uncountable" genes and transcripts is interesting but seems flawed to me. The argument is well made, in particular in the following paragraph, why it could be argued that transcripts are uncountable. But how does this argument apply to genes? And, if genes are uncountable, then how does that impact the efforts to annotate genes across the tree of life?

I thank the reviewer for bringing up this issue. Indeed, the text in my previous submission could appear as contradictory. What I believe is that while transcripts may form a transcriptome continuum and therefore they may be intrinsically uncountable, protein coding genes (specifically the coding regions) are well defined, discrete, entities on the genome sequence, and they are, therefore, countable. This is another argument to dedicate efforts to the identification of protein coding genes. I edited the text to make this clear:

“Genes have been long regarded as well bounded, discrete entities--well described by the “beads on a string” metaphor. While this metaphor could still be largely applied to protein coding genes, coding and non-coding transcripts can overlap extensively, often in complex arrangements with fuzzy boundaries that do not necessarily respect co-linearity with the DNA sequence (57)” (paragraph 3 on page 6)

- The notion of continuum between non-coding and coding, p. 5, was proposed in Nature volume 487, pages370-374 (2012).

I also thank the reviewer for this reference. I believe, however, that I was using the term

continuum in a different, more functional, meaning than Carvunis et al. (2023), where it has a more evolutionary meaning. Carvunis et al. and others suggest that many “de novo” proteins coding genes arise from pre-existing lncRNAs that slowly gain protein coding function. I was referring to transcripts that can act as both protein coding and lncRNAs, and the relative role as one or the other could actually form a continuum (i.e. some transcripts would act as protein coding 90% of the time, and as lncRNAs, 10% of the time, etc).. However, the two meanings are compatible, as genes with dual function may represent transient states in the transition from lncRNAs to proteins. I comment on this on the revised version of the manuscript.

“Rather than assuming a binary typology, there is a continuum along the axis from protein coding to non-coding transcription. Transcripts that are efficiently translated, containing long ORFs, would be preferentially classified as protein coding, while inefficiently translated transcripts containing small ORFs would be classified as lncRNAs. These could actually represent the emergence of novel protein coding genes that often originate from ancestral lncRNAs (62). Often the same transcript would play a dual role as both a lncRNA or a protein coding gene depending on the cellular environment or context, as it has been widely documented (63).” (paragraph 5 on page 6).

- Figure 2 shows change in annotation status of different categories of genes over time in GENCODE, but the main text only talks about the current status, and briefly mentions an increase in annotations in 2009 in the captions. However, there are large fluctuations in protein-coding annotations prior to 2009, and there is the sudden emergence of small RNA annotations in 2005 that seem to be worthy of discussion from a historical context. Also, the “v” dates (e.g. 2005 versus 2005v) in the x axis should be clarified - what do they mean?

I apologize, as the figure was not fully explained. The fluctuations in the number of genes prior to 2009 are only apparent. Between 2005 and 2008, the figure includes two different sets of annotation numbers. One corresponds to computational predictions by ENSEMBL, and the other to the manual curation by the VEGA team from the Sanger Institute (these labeled with a “v”). lncRNAs were only included in the manually curated annotation. From 2009 onwards, the two annotations were merged into a single release GENCODE/ENSEMBL. I do believe that this level of detail is unneeded in a perspective article, and I have regenerated the figure, without the data corresponding to the “v” releases. I believe the point of the constant number of protein coding genes, but the steady increase in lncRNAs is better made this way. Small RNAs within GENCODE are predicted by fully computational pipelines, which were introduced in 2005. Changes in the numbers of these genes mostly reflect changes in the pipelines. I am explaining this in the caption of Figure 2.

- Page 5 third paragraph: “transcripts actually overlap extensively, often in complex arrangements that do not necessarily respect co-linearity with the DNA sequence” - citations should be included.

I added an early, but quite influential reference (Gingeras T, Nature 2009).

- Page 5 fourth paragraph: “the same transcript could have a role as both lncRNA or protein coding gene in different conditions or in different cellular compartments” and “the dual function of many transcripts has been widely documented” - Citations to concrete examples of dual coding/noncoding transcripts would be nice (e.g. SPAAR/LINC00961; Spencer et. al 2020).

I thank the reviewer for this citation, which I have included in the revised version of the manuscript.

- Page 7 paragraph 1: "Ideally, sequence independent normalization methods should be developed." - What is sequence independent normalization?

I clarified this in the text. What I meant is that there is a class of normalization methods that are based on capturing, through sequence probes, the transcripts to be enriched or depleted from the sequencing libraries. To design the probes the sequence of these transcripts should be known prior to normalization, which has little utility to annotate novel genomes, as the location of genes is not known before annotation.

“The dynamic range of gene expression poses a challenge to RNAseq based annotation. Unless resources exist to produce extremely deep RNAseq, many lowly expressed transcripts will remain undetected. To access these transcripts in the human genome, methods have been developed that restrict long-read RNA sequencing to targeted genomic loci (26), or that deplete sequencing libraries for highly abundant transcripts (70). These methods, however, cannot be easily applied to the annotation of new genomes, as they depend on prior knowledge of transcript sequences.” (paragraph 3 on page 8)

- Page 7 paragraph 2: "Automatic pipelines, for instance, do not deal well with exceptions to the universal genetic code." There are other cases that automatic pipelines do not deal well with that may be good to note, such as the identification of shorter coding sequences

I included a mention to the difficulty of the identification of short genes (as well as intronless, antisense, chimeric genes) in this paragraph:

“Expert curation may also be important to deal with other transcript types, which may be also challenging for automatic pipelines. This includes short, intronless, antisense, chimeric transcripts, and, in general g transcripts with low evolutionary sequence conservation, as well as pseudogenes. Expert community curation is particularly important for individual species and gene families to correct errors in automatic predictions and prevent them from propagating across genomes (80). There is even an opportunity to engage the broad scientific community, and even the society as a whole, in genome annotation —for instance, through student-driven community annotation projects (81)”. (paragraph 2 on page 9)

- Page 9 paragraph 1: "Genetic relatedness is likely to have a much larger confounding effect in across-species than in within-species comparisons, as the genomic space is also much larger and, in consequence, it is the range of genetic similarities between genomes." - This sentence seems to have grammatical errors? I am not sure of the meaning but indeed the importance of phylogeny awareness in predicting genotype-phenotype relationships from multi species comparison is well documented.

I meant the following: in GWAS all the genomes belong to individuals of the same species and they are, thus, highly similar. In comparison, genomes from different species show a much larger degree of divergence. In GWAS, the confounding effect of ancestry is usually controlled using principal components on the genotypes and including the resulting principal components as covariates in subsequent GWAS regression models. This approach is not likely to work in comparisons across species, and therefore phylogenetic informed methods need to be employed to control for shared ancestry. I agree that this is already well documented, but I believe that I already acknowledged this in the first submission of the manuscript.. In any case, I simplified the text and I hope that the message is now clearer.

“Because of the strong impact of shared ancestry, any general framework for genomephenome needs to be phylogenetically aware (98). This is an additional challenge, as it is unclear how well phylogenetic methods scale to thousands of genomes. In particular, such analyses often depend on building multiple sequence alignments, which deteriorate with the

size of the input sequence set (99,100).” (paragraph 4 on page 11)

Reviewer #2:

Comments enter in this field will be shared with the author; your identity will remain anonymous. This overview is essential as there are many misconceptions about the process and difficulties of gene annotation. After the publication of the human genome, many researchers, including myself, thought that computational annotation would largely solve the gene annotation problem. In not human genomics circles, there is a strong emphasis on assembly, often with the assumption that annotation will be "free". This article brings the history, current challenges, and opportunities to a broader audience of researchers who are very much dependent on this work.

I am glad that the reviewer found value in the manuscript.

Discussion of issues around annotating TSS/TES, the value of CAGE, and other methods to identify full-length transcripts would enrich this article.

I agree with the reviewer, and I have included this discussion in the revised version of the manuscript.

“In addition, sequencing libraries should be constructed to capture the entire length of the transcript (from the 5’ to the 3’), as the location of promoters is important for understanding gene regulation. A number of protocols are being developed towards that aim. They are generally based on selecting capped RNAs. These include teloprime (71) and technologies that combine long read RNAseq with CAP trapper protocol (72). This protocol has been widely used for CAGE analysis (73), which has been crucial to characterize the regulatory regions in the human genome (74).” (paragraph 3 on page 8)

For Figure 2, it would be good to explain why the number of annotations went down in the early stages of GENCODE. Discussing this will head off questions and help emphasize the importance of new sequencing data.

I apologize, as the figure was not fully explained. The fluctuations in the number of genes prior to 2009 are only apparent. Note that between 2005 and 2008, the figure includes two different sets of annotation numbers. One corresponds to the computational only predictions by ENSEMBL, and the other to the manual curation by the VEGA team from the Sanger Institute (labeled with a “v”). lncRNAs were only included in the manually curated annotation. From 2009 onwards, the two annotations were merged into a single release GENCODE/ENSEMBL. I do believe that this level of detail is unneeded in a perspective article, and I have regenerated the figure, without the data corresponding to the “v” releases. I believe the point of the constant number of protein coding genes, but the steady increase in lncRNAs is better made this way.

The text in several figures is fuzzy, indicating the use of bitmapped rather than vector graphics images.

I regenerate high quality versions of all the figures. I thank the reviewer for pointing this out.

Reviewer #3: Comments enter in this field will be shared with the author; your identity will remain anonymous.

In the manuscript, Dr. Guigo reviewed the history of gene annotation, the different strategies, and then discuss the challenges for finalizing gene annotation in human genome, as well as generating annotations across different eukaryotic species. Overall, the manuscript, if as a

review, did not cover many of the subjects in sufficient depth, if as a perspective, did not provide enough novel insights. Therefore, I do not think it is suitable for a prominent journal as "Cell Genomics".

This manuscript does not attempt to be an exhaustive review, but rather a perspective on the challenges to annotate the genomes of all eukaryotic species and some proposals to overcome them. To understand the origins of these challenges (and of the proposed solutions), I use a brief historical review that helps to frame them in the appropriate context

Referees' reports, second round of review

Reviewer #1: We thank the author for their thorough efforts at addressing our comments and are very glad that they were found useful. The manuscript no longer suffers from any serious limitation in our opinion, and makes for an inspiring read. We would suggest labelling the Y axis of figure 2 and harmonizing the color scheme between lines and text in Figure 3 for increased clarity.

Reviewer #2: Comments enter in this field will be shared with the author; your identity will remain anonymous.

Reviewer #3: In the revised manuscript, the author has incorporated several suggestions from the referees. I do not have further concerns and consider the manuscript appropriate for publication.

Authors' response to the second round of review

Reviewer #1: We thank the author for their thorough efforts at addressing our comments and are very glad that they were found useful. The manuscript no longer suffers from any serious limitation in our opinion, and makes for an inspiring read. We suggest labelling the Y axis of figure 2 and harmonizing the color scheme between lines and text in Figure 3 for increased clarity.

I am pleased the reviewer find the piece of interest. In the revised version of the manuscript, I included a legend in figure 2 that labels the X axis and I harmonize the color schema between lines and text in Figure 3. Thanks for noticing these issues

Reviewer #3: In the revised manuscript, the author has incorporated several suggestions from the referees. I do not have further concerns and consider the manuscript appropriate for publication.

I am also pleased the reviewer find the manuscript appropriate for publication